# The Prognostic Value of Ultrasound Findings in Preoperatively Distinguishing between Uncomplicated and Complicated Types of Pediatric Acute Appendicitis Based on Correlation with Intraoperative and Histopathological Findings

**DOI:** 10.3390/diagnostics12102315

**Published:** 2022-09-26

**Authors:** Konstantina Bekiaridou, Katerina Kambouri, Alexandra Giatromanolaki, Soultana Foutzitzi, Maria Kouroupi, Ioannis Chrysafis, Savas Deftereos

**Affiliations:** 1Department of Pediatric Surgery, Democritus University of Thrace, 68100 Alexandroupolis, Greece; 2Department of Pathology, Democritus University of Thrace, 68100 Alexandroupolis, Greece; 3Department of Radiology, Democritus University of Thrace, 68100 Alexandroupolis, Greece

**Keywords:** ultrasound, complicated appendicitis, uncomplicated appendicitis

## Abstract

Objective: This study compares the preoperative ultrasound findings of all children with a clinical picture of acute appendicitis on the basis of intraoperative and histopathological findings to assess the feasibility of this approach in preoperatively distinguishing between uncomplicated and complicated cases. Methods: This retrospective study includes 224 pediatric patients who underwent ultrasound prior to appendectomy at our institution between January 2016 and February 2022. Logistic regression analysis was used to investigate the association between sonographic and intraoperative histopathological findings. Results: Of the 224 participants, 61.1% were intraoperatively diagnosed with uncomplicated appendicitis (59.8% male). Multivariate logistic regression analysis revealed that patients with a higher appendiceal diameter, presence of appendicolith, and peritonitis were more likely to suffer from complicated appendicitis. Finally, the common anatomical position of the appendix and an appendiceal diameter greater than 6 mm had the highest sensitivity (94.6% and 94.5%, respectively) for predicting complicated appendicitis, with the most specific (99.3%) sonographic finding being the existence of an abscess. Conclusions: Preoperative abdominal ultrasound in children with a clinical diagnosis of acute appendicitis can distinguish between uncomplicated and complicated appendicitis in most cases of pediatric appendicitis. A higher appendiceal diameter, the presence of appendicolith, and peritonitis are parameters noted by ultrasound that strongly predict complicated appendicitis.

## 1. Introduction

Acute appendicitis (AA) in pediatric patients is still the most common cause of acute abdominal pain requiring surgery, with a large proportion not being clinically diagnosed [1]. In the pediatric population, in particular, the percentage not diagnosed ranges from 28% to 57% for patients aged two to twelve years and reaches 100% in the age group of <2 years old [2]. Many aspects of the management of AA remain controversial; as a result, it is a topic of growing interest to pediatric surgeons. Thus, the literature is continuously expanding with different studies on this subject [1].

AA is defined as acute inflammation of the appendix and can occur in patients experiencing acute abdominal pain without any history of appendectomy. The diagnosis of AA should be made as quickly as possible from the onset of symptoms, as the possibility of rupture and development of peritonitis increase with time [3]. The stages of AA, from initial inflammation to the most advanced, have been described with various names, and there are also reports of chronic or relapsing AA. The tendency in recent years is to preoperatively distinguish the simple/uncomplicated cases of appendicitis (inflammatory appendicitis) from the complicated (gangrenous, presence of an appendicolith, abscess, local or generalized peritonitis, or peritoneal mass). The goal is either early surgery to prevent rupture or conservative treatment to reduce the risk of unnecessary surgery [1,4].

The diagnosis of AA in pediatric patients is essentially clinical, based initially on a thorough medical history and clinical examination. However, any clinician should consider that the classic symptomatology of AA in children is usually the exception rather than the rule, as its clinical presentation, as well as its differential diagnosis, is highly dependent on the patient’s age. As the patient gets older, early diagnosis of acute appendicitis becomes more straightforward and accurate [2].

In the pediatric population, abdominal X-ray is still an important tool not only for the exclusion of other causes of pain but also as an initial imaging method for AA diagnosis, but ultrasonography (US) and computed tomography (CT) are considered the imaging methods of choice for this condition [5]. Magnetic resonance imaging (MRI) is an excellent diagnostic method but has limited availability, with the disadvantages of sedation needed in children as it has relatively long duration [6,7]. Moreover, the diagnostic accuracy of US is increasing due to the specialization of the doctors and the evolution of equipment over the years. However, in the case of imaging, the correct diagnosis is highly dependent on the experience of the radiologist performing the US examination [2,3,8].

Accurate US findings should allow preoperatively differentiating between uncomplicated and complicated cases of AA. If this is possible, the surgeon can then safely decide whether it is feasible to use conservative treatment, the timing of the surgery, how to reduce complications, and how to avoid an unnecessary operation. The present study aims to evaluate US as a useful and discriminatory tool to differentiate between uncomplicated and complicated appendicitis. 

## 2. Materials and Methods

### 2.1. Study Population

In the present study, we retrospectively examined the records of patients aged 0–14 years who were hospitalized in the Pediatric Surgery Department of Alexandroupolis University Hospital, Democritus University of Thrace. The study included children who had preoperative abdominal US and had an operation for AA between January 2016 and February 2022. Finally, the preoperative US findings were compared with the intraoperative histopathological findings after appendectomy. Exclusion criteria for our study were the absence of any data related to the above parameters in the patient’s record as well as cases of secondary or elective appendectomy, histopathologically confirmed normal appendix, and carcinoid or other pathology. In total, 27 out of 251 children with a clinical picture of AA who were preoperatively examined by abdominal US and underwent an operation were excluded from further analysis. Specifically, patients were excluded due to missing data (*n* = 14), secondary or elective appendectomy (*n* = 5), histopathologically confirmed carcinoid or other pathology (*n* = 3), and negative histopathology (*n* = 5). The study was conducted according to the guidelines of the Declaration of Helsinki, and the original protocol was approved by the Medical Ethics Committee of Alexandroupolis University Hospital (approval number 6809/19-02-2021).

### 2.2. Intraoperative Classification

Intraoperative recorded findings (IF) were retrospectively reviewed to classify appendicitis. The following two major types of appendicitis were differentiated: acute uncomplicated appendicitis (AUA) and acute complicated appendicitis (ACA). As stated in the literature, AUA was defined as inflammatory appendicitis without gangrene or perforation, while ACA was defined as gangrenous appendicitis with or without perforation, the presence of abscess or an appendicolith, and local or generalized peritonitis due to appendix rupture [9,10,11,12].

### 2.3. Histopathological Classification

Histopathological diagnoses were retrospectively reviewed to classify appendicitis. The following three major types of appendicitis were differentiated: normal appendix (NA); inflammatory appendicitis (IA) without gangrene, appendicolith, or perforation; and gangrenous appendicitis (GA) with or without perforation, abscess, or the presence of an appendicolith. As stated in the literature, an appendix without significant histopathological changes was defined as minimal changes with mild lymphoplasmacytic infiltration in the mucosa (NA) and was excluded from our study, IA was characterized by the intense presence of neutrophils in all layers, corresponding to AUA, and GA with extensive inflammatory infiltration mainly composed of neutrophils and lymphocytes involving thickness of the appendiceal wall, with/or without appendicolith, with/or without perforation, defined by the presence of a transmural defect, corresponding to ACA [13,14,15].

### 2.4. Ultrasound Examination

The US findings were examined by general radiologists and were re-evaluated by an experienced pediatric radiologist having full access to the medical files of all eligible patients. Abdominal US was performed with a convex or linear transducer (5, 8, and 12–17 MHZ, respectively) adapted to the patient’s body constitution. In all patients, the US method was that of slowly applied intermittent compression, which is ideal to diminish the patient’s reaction as well as to allow compression of the ventrally located bowel loop, which reduces the negative influence of gas and enables easier identification of the appendix and/or appendicitis. Using this well-known method, the following findings were recorded: aperistaltic noncompressible appendix (appears around when compression is applied); appendiceal lumen diameter (dilated appendix > 6 mm, measured from outer wall to outer wall in mm); anatomical position of the appendix; distinct appendiceal wall layers (preservation of normal appendiceal wall layers echomorphology or obliteration of these layers which defined as a high chance for perforation); target appearance (in axial section); inflammation of the periappendiceal fat (defined as an increased echogenicity of the periappendiceal fat); the presence of free intraperitoneal fluid as periappendiceal fluid collection or diffuse with or without internal echogenicities, which constitute a high ultrasonographic suspicion of peritonitis; the presence of appendicolith within the appendix (as an intraluminal hyperechogenic formation with an acoustic shadow); lymphadenitis (sonographically detectable lymph nodes in proximity to the appendix or diffuse but usually as periappendiceal reactive nodal enlargement); the presence of an abscess or peritoneal mass (walled-off mass grouped together with appendix indistinguishable from other intestinal structures); mural hyperemia with color flow Doppler; and loss of vascular flow in necrotic stages. Finally, to ensure that the visualized structure was the appendix, it was confirmed as blind-ending and arising from the base of the cecum. The decision to include the above US findings in our study was based on the most recent data from previous studies through a systematic search of the Medline literature [2,16].

All US findings were compared to the intraoperative/histopathological findings, which was used as our gold standard test. The correlation of the above is shown in Figure 1.

### 2.5. Statistical Analysis

Quantitative variables were presented as mean (SD) values. Qualitative variables were expressed as absolute and relative frequencies (N, %). For comparison of proportions, chi-square and Fisher’s exact tests were used. Univariate logistic regression analysis was applied to explore the US findings associated with intraoperative/histopathological diagnosis of ACA. A backward model selection method was used to obtain the final multivariate model. The variable with the largest *p*-value was removed from the model each time and only covariates that were significant at *p* < 0.10 were kept in the final model. Odds ratios with 95% confidence intervals were computed from the results of the logistic regression analyses. The sensitivity, specificity, and positive and negative predictive values were estimated for each US finding. All the statistical analyses were performed using STATA 16.0. Values of *p* < 0.05 are considered to indicate statistically significant differences.

## 3. Results

A total of 224 subjects took part in the study. As shown in Table 1, most of the participants were male (59.8%) with a mean (SD) age of 9.41 (2.69) years. Patients suffering from ACA were significantly younger than those suffering from AUA (8.66 vs. 9.87, *p* = 0.005). The mean appendiceal diameter was significantly higher in patients with ACA (10.27 vs. 8.32, *p* < 0.001). Most of the patients had the usual anatomical position (79.9%). A significantly higher percentage of ACA patients had distinct appendiceal wall layers (52.3% vs. 31.9%, *p* = 0.001). In total, 82.6% of the patients had a noncompressible appendix. More patients with AUA had an appearance of a target sign (63.8% vs. 36%, *p* < 0.001). Most of the patients had no hypervascularization (85.3%). The presence of appendicolith was associated with higher rates of patients with ACA (36% vs. 13.8%, *p* < 0.001). Periappendiceal fat inflammation and free intraperitoneal fluid (FIF) were observed in most patients with ACA (61.6% vs. 44.2%, *p* = 0.004; 70.9% vs. 55.8%, *p* = 0.024, respectively). Similar results were also found for diffuse free intraperitoneal fluid (DFIF) (20.9% vs. 9.4%, *p* = 0.015). Lymphadenitis was observed in 34.8% of the patients. Finally, abscess (14% vs. 0.7%, *p* < 0.001) and peritonitis (14% vs. 1.4%, *p* < 0.001) were more common in patients with ACA. No statistically significant differences were found in terms of sex, anatomical position, noncompressible appendix, hypervascularization, free intraperitoneal fluid in the periappendiceal region (PFIF), free intraperitoneal fluid in Douglas’s pouch (DPFIF), and lymphadenitis between two groups.

According to the univariate logistic regression models, the factors associated with ACA were age as continuous variables (OR = 0.84, *p* = 0.001) and as binary (OR = 0.51, *p* = 0.017), appendiceal diameter (OR = 1.30, *p* < 0.001), distinct appendiceal wall layers (OR = 2.86, *p* = 0.001), target sign appearance (OR = 0.30, *p* < 0.001), appendicolith (OR = 4.02, *p* < 0.001), periappendiceal fat inflammation (OR = 2.65, *p* = 0.002), free abdominal fluid (OR = 1.93, *p* = 0.024), DFIF (OR = 2.55, *p* = 0.018), PFIF (OR = 1.83, *p* = 0.042), abscess (OR = 22.22, *p* = 0.003) and peritonitis (OR = 11.03, *p* = 0.002). The multivariate logistic regression analysis with age as a continuous variable revealed that older patients (OR = 0.83, *p* = 0.013) and patients who had a target sign appearance (OR = 0.29, *p* = 0.001) were less likely to suffer from ACA. On the contrary, patients with a higher appendiceal diameter (OR = 1.32, *p* <0.001), appendicolith (OR = 3.75, *p* < 0.001) or peritonitis (OR = 12.50, *p* = 0.008) were more likely to suffer from ACA. An additional analysis was performed, introducing age with two categories in the logistic model. The results showed that patients aged 10–15 years (OR = 0.34, *p* = 0.005) were less likely to suffer from ACA compared to patients aged 0–10 years. In addition, patients who had target sign appearance (OR = 0.24, *p* < 0.001) were less likely to suffer from ACA. On the contrary, patients with a higher appendiceal diameter (OR = 1.34, *p* < 0.001), with usual anatomical position (OR = 4.62, *p* = 0.039), with appendicolith (OR = 4.05, *p* = 0.001) and peritonitis (OR = 13.56, *p* = 0.005) were more likely to suffer from ACA (Table 2).

The sensitivity of the usual anatomical position in predicting ACA was 94.6%, which was the highest value. In contrast, the specificity of this characteristic was very low (12.8%). Moreover, the sensitivity of an appendiceal diameter greater than 6 mm for the detection of ACA was 94.5%, quite close to the sensitivity of the anatomical position, but the specificity of the appendiceal wall diameter was lower (7.2%). The most specific (99.3%) US finding of ACA was the existence of an abscess (Table 3).

## 4. Discussion

Preoperatively distinguishing between the different forms of AA using US screening by radiologists will be particularly useful in the coming years, as several epidemiological and immunological studies have recently supported that histopathologically confirmed inflammatory appendicitis (corresponding to the clinical form of uncomplicated appendicitis (AUA)) and gangrenous appendicitis possibly leading to rupture (clinically corresponding to complicated appendicitis (ACA)) appear to have different origins and may need different treatments [9].

Histopathological examination remains the gold standard for confirming the diagnosis of AA, regardless of the intraoperative findings [17,18]. In the present study, five patients had negative histopathology (2.23%) in contrast to the intraoperative findings for AUA and three patients were histopathologically confirmed to have other pathologies; therefore, all were excluded from our study. The presence of a percentage of negative histopathology can be attributed to differences in the experience of each pediatric surgeon and their ability to recognize and describe each intraoperative finding, in addition to the lack of a universally accepted system for classification of intraoperative findings. Although, in our study, there was a very small percentage of negative histopathology, we preferred to exclude these cases from the overall results [17,18,19,20,21].

It has been reported that the use of MRI and CT in preoperative diagnosis of AA in AUA and ACA has high accuracy; in particular, the distinction of a ruptured appendix has great specificity (85–99%) with these methods [9]. However, in the pediatric population, there is significant concern regarding increased radiation exposure and long-term cancer risks associated with using CT. Moreover, other studies refer to the difficulty children have in receiving intravenous contrast and sedation, and others recognize that CT or MRI are not particularly effective at diagnosing AUA or ACA [22,23,24]. There are very few studies that distinguish AUA and ACA using US, which motivated us in conducting this study and served as guidance in establishing the details of the study design [9,13].

This distinction by US examination is important for many reasons. The early imaging recognition of gangrenous appendicitis with or without rupture can lead to selecting the right time for initiation of antibiotic treatment as well as an early surgical intervention to reduce the effects and complications of peritonitis. Moreover, although AA is a surgical condition for which surgical removal of the appendix is the gold standard treatment, high rates of postoperative complications have recently been reported [9,25]. Appendicitis is also a disease of many presentations; a part of AA cases is reversible and will heal without any interventions [20]. This has resulted in several studies recommending conservative treatment of AA in its early uncomplicated stages as acceptable [10,26]. Hence, the use of US in combination with clinical imagery for the preoperative differentiation between AUA and ACA will be important for determining and categorizing the forms of AA and, therefore, the appropriate specific treatment to be applied [27].

Regarding the US findings, it was noted that, in 25 of our patients (11.2%), it was not possible to identify the appendix (N/I). From those, 13 were diagnosed with AUA and 12 with ACA according to intraoperative histopathological classification. This may be due to the lack of expertise of our hospital’s radiologists in pediatric radiology (there is one pediatric radiologist who cannot be on call every time a pediatric patient needs an abdominal US), the possibility of child obesity, or the retrocecal location of the appendix [16,28]. Although the expert pediatric radiologist evaluated all imaging results in these occasions, it should be kept in mind that US is a real-time/dynamic examination and the images are not often reliable when reviewed retrospectively. Our results of non-identified appendicitis via US indicated very good performance in comparison to the literature, where the rates of appendix nonvisualization by US ranged from 20 to 75% [29,30]. The visualization rate seems to improve after clinical re-evaluation, and a second US with high awareness of secondary signs (periappendiceal fat inflammation, free fluid, appendicolith, abscess, and lymphadenitis) tends to lead to the correct diagnosis [31,32].

In our study, statistically significant differences were frequently observed in many US findings showing that some images were clearly characteristic of either ACA or AUA. A significantly higher percentage of ACA patients had distinct appendiceal wall layers, the presence of appendicolith, periappendiceal fat inflammation, free intraperitoneal fluid (FIF), and finally, abscess and peritonitis, whereas more patients with AUA had an appearance of the target sign. According to the multivariate analysis in our study, it was confirmed that patients with a higher appendiceal diameter (greater than 6 mm), appendicolith, and peritonitis were significantly associated with ACA, while patients who were older and had target sign appearance were less likely to suffer from ACA. Our data are in agreement with the results of previously published studies [33,34]. However, the existence of an appendicolith can be observed in both normal and abnormal appendices, and it remains controversial whether an appendicolith leads to an increased risk of appendicitis [35]. Nevertheless, in our study, the combination of the appendicolith, signs of peritonitis, and greater appendiceal diameter predicted not only the existence of acute appendicitis but that it was of ACA type.

On the other hand, it was observed in our study that older children with existence of the target sign of the appendix in US and not irregularity of the appendiceal wall layers were less likely to suffer from ACA. The loss in compressibility or the fact that the appendix adopted a circular shape in axial images provides this target sign image that is characteristic in the early stages of AA. In contrast, as inflammation progresses, irregularity of the appendiceal wall is often observed, or the appendix commonly remains unidentified because of perforation [36].

The dominant appendix position observed in our study was the usual anatomical position (79.9%), including for patients suffering from ACA (81.4%). There is a controversy in the literature about the link between appendix location and disease severity. Some studies state that an appendix in an abnormal location is likely to have an atypical clinical presentation, resulting in a higher incidence of misdiagnoses and complications, while others state that the position of the appendix does not alter the symptomatology [37,38]. Our study indicates that the appendix’s typical and not atypical position is associated with a higher incidence of ACA. Moreover, the typical position of the appendix in our study had great sensitivity for ACA (94.6%).

Another interesting finding of our study was the association of the presence of free intraperitoneal fluid in patients with ACA, mainly in its diffuse form or in a periappendiceal location, as determined by our univariate analysis, which is also supported by reports in the literature [33,34,39]. Although the present finding was not evaluated as satisfactory in our multivariate analysis, possibly due to the number of patients in our study, further research is needed to conclude if the location of fluid is correlated with ACA or if it is a finding that is not associated with children with a clinical picture of AA.

Another point of argument in the literature is the appendiceal diameter. Some authors state that up to 23% of adult males have a cecal-healthy appendix with a diameter larger than 6 mm. Therefore, they suggest that when an appendix with a diameter between 6 and 9 mm is found, it must be considered “undetermined”, and other signs of appendicitis must be searched for [40]. In our study, an appendiceal diameter greater than 6 mm had 94.5% sensitivity for the detection of ACA, meaning that this diameter is a characteristic of the complicated form of AA.

Finally, the existence of an abscess proved to be the most specific sonographic finding of ACA (99.3%) in our study. Although there are several studies confirming the association of US signs of peritonitis with or without the presence of a periappendiceal abscess with ACA, few have directed their main focus on the latter. This is likely due to its low incidence in the general population (2–7%) [41,42].

The limitations of our study were that it was retrospective and, therefore, more specific US findings could not be designed to examine in every US for detection of AA. We did not have many specialized radiologists available to examine pediatric appendixes with US in real time, and we did not have a large number of cases to ensure the reliability of our results. However, it is crucial to mention that there are very few studies that compare US findings with histopathological and intraoperative findings of AA to demonstrate the reliability of sonographic findings as well as the points on which a radiologist may focus to distinguish whether the AA is simple or complicated.

## 5. Conclusions

Abdominal US, conducted preoperatively by radiologists, and if possible, by pediatric radiologists, can distinctly illustrate the possible presence of either AUA or ACA in pediatric patients. A higher appendiceal diameter, the presence of appendicolith, and peritonitis are parameters determined by US that are highly predictive of ACA. Furthermore, the confidence of this diagnosis can be potentially strengthened by supporting data on the anatomical position of the appendix as well as the diffuse and periappendiceal presence of free intraperitoneal fluid, but further research and more replication studies will be needed to verify these results and possibly use them in the treatment algorithm in the future.

## Figures and Tables

**Figure 1 diagnostics-12-02315-f001:**
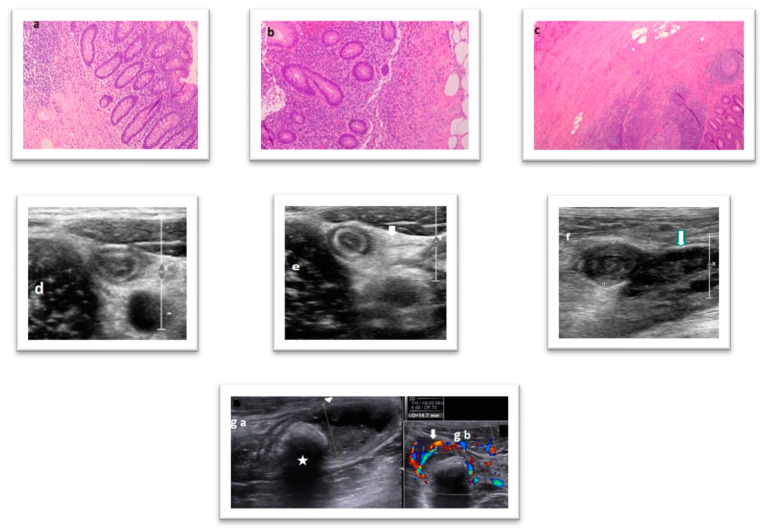
Correlation between US and intraoperative/histopathological findings. (**a**) Normal appendix: minimal changes—mild lymphoplasmacytic infiltration in the mucosa (H&E stain, magnification ×10). (**b**) Acute uncomplicated appendicitis: intense presence of neutrophils (H&E stain, magnification ×10). (**c**) Acute complicated appendicitis without perforation: extensive inflammatory infiltration mainly composed of neutrophils and lymphocytes involving all thickness of the organ’s wall (H&E stain, magnification ×4). (**d**) Normal compressible appendix (note the ovoid shape because of the compression). (**e**) Uncompressible, with “target sign” appendix in axial view. Uncomplicated AA but with increased echogenicity of periappendiceal fat (arrow), which represents an attempt to wall-off the imminent perforation. (**f**) Thickening and ill-defined “target-sign” in the edematous appendix. Periappendiceal hypoechoic fluid collection (arrow) with internal echoes is present [complicated AA]. (**g**) Appendicolith with acoustic shadow [star-(**ga**)] obstructing the appendiceal lumen (diameter approximately 15 mm). Periappendiceal fat and mural hypervascularity is present [arrow in (**gb**)] and obliteration of wall layer structure indicates a high chance for perforation [complicated AA].

**Table 1 diagnostics-12-02315-t001:** Distribution of age, sex and US findings in total sample and by histopathological classification.

Characteristic	Total (N = 224)	ACA (N = 86)	AUA (N = 138)	*p* *
**Age (years)**	9.41 (2.69)	8.66 (3.02)	9.87 (2.36)	**0.005**
**Age categories (years)**				**0.016**
**0–** **10**	110 (49.1%)	51 (59.3%)	59 (42.8%)	
**10–15**	114 (50.9%)	35 (40.7%)	79 (57.2%)	
**Appendiceal diameter(mm)**	9.04 (2.88)	10.27 (3.52)	8.32 (2.13)	**<0.001**
**Gender**				0.663
Male	134 (59.8%)	53 (61.6%)	81 (58.7%)	
Female	90 (40.2%)	33 (38.4%)	57 (41.3%)	
**Anatomical position**				0.144
Usual	179 (79.9%)	70 (81.4%)	109 (79%)	
Unusual	20 (8.9%)	4 (4.7%)	16 (11.6%)	
N/I	25 (11.2%)	12 (14%)	13 (9.4%)	
**Distinct appendiceal** **Wall** **layers**				**0.001**
Yes	89 (39.7%)	45 (52.3%)	44 (31.9%)	
No	110 (49.1%)	29 (33.7%)	81 (58.7%)	
N/I	25 (11.2%)	12 (14%)	13 (9.4%)	
**Non-compressible**				0.688
Yes	185 (82.6%)	70 (81.4%)	115 (83.3%)	
No	15 (6.7%)	5 (5.8%)	10 (7.2%)	
N/I	24 (10.7%)	11 (12.8%)	13 (9.4%)	
**Target sign appearance**				**<0.001**
Yes	119 (53.1%)	31 (36%)	88 (63.8%)	
No	80 (35.7%)	43 (50%)	37 (26.8%)	
N/I	25 (11.2%)	12 (14%)	13 (9.4%)	
**Hypervascularisation**				0.197
Yes	8 (3.6%)	1 (1.2%)	7 (5.1%)	
No	191 (85.3%)	73 (84.9%)	118 (85.5%)	
N/I	25 (11.2%)	12 (14%)	13 (9.4%)	
**Appendicolith**				**<0.001**
Yes	50 (22.3%)	31 (36%)	19 (13.8%)	
No	149 (66.5%)	43 (50%)	106 (76.8%)	
N/I	25 (11.2%)	12 (14%)	13 (9.4%)	
**Periappendiceal fat inflammation**				**0.004**
Yes	114 (50.9%)	53 (61.6%)	61 (44.2%)	
No	85 (37.9%)	21 (24.4%)	64 (46.4%)	
N/I	25 (11.2%)	12 (14%)	13 (9.4%)	
**Free abdominal fluid**				**0.024**
Yes	138 (61.6%)	61 (70.9%)	77 (55.8%)	
No	86 (38.4%)	25 (29.1%)	61 (44.2%)	
**DFIF**				**0.015**
Yes	31 (13.8%)	18 (20.9%)	13 (9.4%)	
No	193 (86.2%)	68 (79.1%)	125 (90.6%)	
**PFIF**				0.093
Yes	83 (37.1%)	38 (44.2%)	45 (32.6%)	
No	117 (52.2%)	37 (43%)	80 (58%)	
N/I	24 (10.7%)	11 (12.8%)	13 (9.4%)	
**DPFIF**				0.448
Yes	53 (23.7%)	18 (20.9%)	35 (25.4%)	
No	171 (76.3%)	68 (79.1%)	103 (74.6%)	
**Lymphadenitis**				0.761
Yes	78 (34.8%)	31 (36%)	47 (34.1%)	
No	146 (65.2%)	55 (64%)	91 (65.9%)	
**Abscess**				**<0.001**
Yes	13 (5.8%)	12 (14%)	1 (0.7%)	
No	211 (94.2%)	74 (86%)	137 (99.3%)	
**Peritonitis**				**<0.001**
Yes	14 (6.3%)	12 (14%)	2 (1.4%)	
No	210 (93.8%)	74 (86%)	136 (98.6%)	

Note: AUA: acute uncomplicated appendicitis, ACA: acute complicated appendicitis, N/I: not identified, DFI: diffuse free intraperitoneal fluid, PFIF: free intraperitoneal fluid in the periappendiceal region, DPFIF: free intraperitoneal fluid in Douglas’s pouch.* refers to the statistical tests performed to compare ACA vs. AUA.

**Table 2 diagnostics-12-02315-t002:** Logistic regression analysis for ACA and US findings.

	Univariate Logistic Regression		Multivariate Logistic Regression *		Multivariate Logistic Regression **	
	OR (95%CI)	*p*	OR (95%CI)	*p*	OR (95%CI)	*p*
Gender, Male	1.13 (0.65–1.96)	0.663				
Age (years)	0.84 (0.76–0.94)	**0.001**	0.83 (0.72–0.96)	**0.013**		
Age categories, 10–15 years	0.51 (0.30–0.88)	**0.017**			0.34 (0.16–0.72)	**0.005**
Appendiceal diameter (mm)	1.30 (1.15–1.48)	**<0.001**	1.32 (1.13–1.55)	**<0.001**	1.34 (1.14–1.57)	**<0.001**
Anatomical position, Usual	2.57 (0.82–8)	0.104	3.78 (0.92–15.52)	0.065	4.62 (1.08–19.82)	**0.039**
Anatomical position, N/I	3.69 (0.96–14.21)	0.057				
Distinct appendiceal wall layers, Yes	2.86 (1.58–5.17)	**0.001**				
Distinct appendiceal wall layers, N/I	2.58 (1.06–6.29)	**0.037**				
Non-compressible, Yes	1.22 (0.4–3.71)	0.729				
Non-compressible, N/I	1.69 (0.44–6.47)	0.442				
Target sign appearance, Yes	0.30 (0.17–0.55)	**<0.001**	0.29 (0.14–0.6)	**0.001**	0.24 (0.11–0.51)	**<0.001**
Target sign appearance, N/I	0.79 (0.32–1.95)	0.616				
Hypervascularisation, Yes	0.23 (0.03–1.92)	0.174				
Hypervascularisation, N/I	1.49 (0.65–3.45)	0.349				
Appendicolith, Yes	4.02 (2.05–7.88)	**<0.001**	3.75 (1.7–8.24)	**0.001**	4.05 (1.81–9.05)	**0.001**
Appendicolith, N/I	2.28 (0.96–5.38)	0.061				
Periappendiceal Fat inflammation, Yes	2.65 (1.43–4.9)	**0.002**				
Periappendiceal fat inflammation, N/I	2.81 (1.11–7.11)	**0.029**				
Free abdominal fluid, Yes	1.93 (1.09–3.43)	**0.024**				
DFIF, Yes	2.55 (1.18–5.51)	**0.018**				
PFIF, Yes	1.83 (1.02–3.27)	**0.042**				
PFIF, N/I	1.83 (0.75–4.47)	0.185				
DPFIF, Yes	0.78 (0.41–1.49)	0.448	0.43 (0.17–1.06)	0.066	0.4 (0.16–1.01)	0.052
Lymphadenitis, Yes	1.09 (0.62–1.92)	0.761				
Abscess, Yes	22.22 (2.83–174.22)	**0.003**				
Peritonitis, Yes	11.03 (2.4–50.59)	**0.002**	12.5 (1.95–80.18)	**0.008**	13.56 (2.16–85.2)	**0.005**

Note: OR = odds ratio, CI = confidence interval, *p* < 0.05, * age as continuous variable, ** age as binary variable.

**Table 3 diagnostics-12-02315-t003:** Sensitivity and specificity values of US findings for ACA.

Sonographic Finding	Sensitivity (%)	Specificity (%)	PPV (%)	NPV (%)
Appendiceal Wall diameter (mm)				
≤5	4.1 (0.9, 11.5)	96.8 (92, 99.1)	42.9 (9.9, 81.6)	63.4 (56.1, 70.2)
≥6	**94.5 (86.6, 98.5)**	**7.2 (3.3, 13.2)**	**37.3 (30.3, 44.7)**	**69.2 (38.6, 90.9)**
≥8	78.1 (66.9, 86.9)	48.8 (39.8, 57.9)	47.1 (38, 56.4)	79.2 (68.5, 87.6)
≥10	47.9 (36.1, 60)	80.8 (72.8, 87.3)	59.3 (45.7, 71.9)	72.7 (64.5, 79.9)
≥12	24.7 (15.3, 36.1)	92 (85.8, 96.1)	64.3 (44.1, 81.4)	67.6 (60.1, 74.6)
**Anatomical position**	**94.6 (86.7, 98.5)**	**12.8 (7.5, 20)**	**39.1 (31.9, 46.7)**	**80 (56.3, 94.3)**
Distinct Appendiceal Wall layers	60.8 (48.8, 72)	64.8 (55.8, 73.1)	50.6 (39.8, 61.3)	73.6 (64.4, 81.6)
Non-compressible	93.3 (85.1, 97.8)	8 (3.9, 14.2)	37.8 (30.8, 45.2)	66.7 (38.4, 88.2)
Target sign appearance	41.9 (30.5, 53.9)	29.6 (21.8, 38.4)	26.1 (18.4, 34.9)	46.3 (35, 57.8)
Hypervascularisation	1.4 (0, 7.3)	94.4 (88.8, 97.7)	12.5 (0.3, 52.7)	61.8 (54.5, 68.7)
Appendicolith	41.9 (30.5, 53.9)	84.8 (77.3, 90.6)	62 (47.2, 75.3)	71.1 (63.2, 78.3)
Periappendiceal Fat inflammation	71.6 (59.9, 81.5)	51.2 (42.1, 60.2)	46.5 (37.1, 56.1)	75.3 (64.7, 84)
Free abdominal fluid	70.9 (60.1, 80.2)	44.2 (35.8, 52.9)	44.2 (35.8, 52.9)	70.9 (60.1, 80.2)
DFIF	20.9 (12.9, 31)	90.6 (84.4, 94.9)	58.1 (39.1, 75.5)	64.8 (57.6, 71.5)
PFIF	50.7 (38.9, 62.4)	64 (54.9, 72.4)	45.8 (34.8, 57.1)	68.4 (59.1, 76.7)
DPFIF	20.9 (12.9, 31)	74.6 (66.5, 81.7)	34 (21.5, 48.3)	60.2 (52.5, 67.6)
Lymphadenitis	36 (26, 47.1)	65.9 (57.4, 73.8)	39.7 (28.8, 51.5)	62.3 (53.9, 70.2)
**Abscess**	**14 (7.4, 23.1)**	**99.3 (96, 100)**	**92.3 (64, 99.8)**	**64.9 (58.1, 71.4)**
Peritonitis	14 (7.4, 23.1)	98.6 (94.9, 99.8)	85.7 (57.2, 98.2)	64.8 (57.9, 71.2)

## Data Availability

The datasets generated during and/or analyzed during the current study are available from the corresponding author on reasonable request.

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
