# Peer review of "The Prognostic Value of Ultrasound Findings in Preoperatively Distinguishing between Uncomplicated and Complicated Types of Pediatric Acute Appendicitis Based on Correlation with Intraoperative and Histopathological Findings"

_diagnostics, 2022, doi:10.3390/diagnostics12102315_

Round 1

Reviewer 1 Report

Acute appendicitis still poses diagnostic challenges, especially in children. Delays in diagnosis increase the risk of morbidity and related complications. Thus, any high-quality reports, which aim to improve diagnostics, are most welcome. The current manuscript analyses acute appendicitis and compares preoperative us findings with intraoperative and histopathological findings. The study setup is appropriate.

A few comments:

-          the use of professional English editing is recommended as the language is occasionally awkward and incorrect. Although the intentions of the authors are vaguely identified, the message is lost in all the language faults.

-          line 63 in introduction states that abdominal x-ray is the initial imaging method for AA in the pediatric population, which most certainly is not true. The authors most probably intend to say that radiological methods of US and CT are the method of choice, but the way the sentence is written currently states that abdominal x-ray would be the initial imaging which has to be corrected.

-          Could the authors subgroup their patients into age categories? As they state, factors associated with complicated AA are age, appendiceal diameter, distinct appendiceal wall layers, target sign, appendicolith, periappenciceal fat inflammation, free intraperitoneal fluid, abscess and peritonitis. How are these findings correlated to presenting age of the patient? After all, it is of interest to attain a correct diagnosis early especially in younger children, who may at times be unable to verbally aid reaching the diagnosis.

-          it was not clear what was included in complicated appendicitis. While complicated appendicitis definitely entails perforated disease, it is not limited to that end. It is customary to consider AA in histopathology as phlegmonous vs gangrenous, but is that the best way needs to be addressed in the Discussion. The etiology of these also needs to be addressed, as well as how to distinguish them clinically given the results from this study.

-          While the Introduction is acceptable, the Discussion fails to complement it. The Discussion fails especially due to language problems, but also in not providing a broader view on diagnostics of appendicitis per se. Given the current evidence regarding diagnostic accuracy, clinical vigilance remains the mainstay for proper treatment choices in pediatric AA, complemented by radiological AND biochemical methods and I think it would be of utmost importance to stress this.

-          In the Discussion (line 250) the authors state that AA is a surgical condition seems a bit narrow-sighted. Antimicrobial therapy has been deemed safe and cost-effective in patients with confirmed uncomplicated appendicitis. It is not the rates of postoperative complications that suggest that part of the disease should be treated conservatively, rather it is to be acknowledged that appendicitis in itself is a disease of many presentations. A part of AA cases are reversible and will heal without any interventions.

-          While I think that this manuscript has much potential, especially the Discussion needs revision INCLUDING editing of the English language.

Author Response

Reviewer 1 comments: Comments and Suggestions for Authors

Acute appendicitis still poses diagnostic challenges, especially in children. Delays in diagnosis increase the risk of morbidity and related complications. Thus, any high-quality reports, which aim to improve diagnostics, are most welcome. The current manuscript analyses acute appendicitis and compares preoperative us findings with intraoperative and histopathological findings. The study setup is appropriate.

A few comments:

-          the use of professional English editing is recommended as the language is occasionally awkward and incorrect. Although the intentions of the authors are vaguely identified, the message is lost in all the language faults.

  Thank you very much for your valuable comments. We made any effort  to make some changes and extended English editing ( MDPI editing service)

-          line 63 in introduction states that abdominal x-ray is the initial imaging method for AA in the pediatric population, which most certainly is not true. The authors most probably intend to say that radiological methods of US and CT are the method of choice, but the way the sentence is written currently states that abdominal x-ray would be the initial imaging which has to be corrected.

 We changed the phrase about xray, and added the relevant reference.

-          Could the authors subgroup their patients into age categories? As they state, factors associated with complicated AA are age, appendiceal diameter, distinct appendiceal wall layers, target sign, appendicolith, periappenciceal fat inflammation, free intraperitoneal fluid, abscess and peritonitis. How are these findings correlated to presenting age of the patient? After all, it is of interest to attain a correct diagnosis early especially in younger children, who may at times be unable to verbally aid reaching the diagnosis.

About the age’s subgroups, initially, a new age variable with three categories: 0–4 years, 4–10 years, and 10–15 years was created and used in the logistic regression analysis. The results were not reliable as patients aged 0-4 years were only 7, while in the other two categories the patients were 103 and 114 respectively. For this reason, the first two categories (1: 0-10 years and 10-15 years) were combined and used in the regression model. The results were presented in Table 2.

-          it was not clear what was included in complicated appendicitis. While complicated appendicitis definitely entails perforated disease, it is not limited to that end. It is customary to consider AA in histopathology as phlegmonous vs gangrenous, but is that the best way needs to be addressed in the Discussion. The etiology of these also needs to be addressed, as well as how to distinguish them clinically given the results from this study.

We describe what are the characteristics in each condition. The following is the paragraph between lines 168-179

Histopathological diagnoses were retrospectively reviewed to classify appendicitis. The following three major types of appendicitis were differentiated: normal appendix (NA); inflammatory appendicitis (IA) without gangrene, appendicolith, or perforation; and gangrenous appendicitis (GA) with or without perforation, abscess, or the presence of an appendicolith. As stated in the literature, an appendix without significant histopathological changes was defined as minimal changes with mild lymphoplasmacytic infiltration in the mucosa (NA) and was excluded from our study, IA was characterized by the intense presence of neutrophils into all layers, corresponding to AUA ,and GA with extensive inflammatory infiltration mainly composed of neutrophils and lymphocytes involving thickness of the appendiceal wall, with/or without appendicolith, with/or without perforation, defined by the presence of a transmural defect, corresponding to ACA-          While the Introduction is acceptable, the Discussion fails to complement it. The Discussion fails especially due to language problems, but also in not providing a broader view on diagnostics of appendicitis per se. Given the current evidence regarding diagnostic accuracy, clinical vigilance remains the mainstay for proper treatment choices in pediatric AA, complemented by radiological AND biochemical methods and I think it would be of utmost importance to stress this.

-          In the Discussion (line 250) the authors state that AA is a surgical condition seems a bit narrow-sighted. Antimicrobial therapy has been deemed safe and cost-effective in patients with confirmed uncomplicated appendicitis. It is not the rates of postoperative complications that suggest that part of the disease should be treated conservatively, rather it is to be acknowledged that appendicitis in itself is a disease of many presentations. A part of AA cases are reversible and will heal without any interventions.

We added a phrase in the discussion section about not only the surgical view of treatment but also the cases that are reversible without interventions as you proposed.

-          While I think that this manuscript has much potential, especially the Discussion needs revision INCLUDING editing of the English language.

We think that the language improved after editing

Reviewer 2 Report

This is a single-centre study, and all the results must be viewed as such – please stress this in the limitations section. English will require moderate improvements. The Abstract should not report univariate results, as they are misleading. On the contrary, you should report ORs and CIs for the multivariate significant predictors. The main question is, why bother with ultrasound if the operation can be performed? There is an additional cost, and the problem is only diagnosed. Of course, it may drive and guide more complicated cases, but it could be argued that this entire procedure is not needed in appendicitis. The sample breakdown is another problem – how many vases of appendicitis did you operate that did not have the pre-operative US? Change the less or equal P to 0.05 to only less than 0.05. Downplay the univariate results; they are misleading. Explain the ACA and AUA abbreviations again in the header of Table 1. Explain what P stands for in Table 1, ACA vs all, AUA vs all or ACA vs AUA? However I checked the chi-square results, they are not the same as you showed them. Please verify. The discussion aligns well and restores some of my previous questions, so it fits with the paper scope nicely. Please suggest more replication studies so these results can be verified and possibly used in the treatment algorithm in the future. Please mention the fact that you did not use the live US, but a retrospective US result in the diagnostics; this can have a negative effect, as the US requires a lot of manipulation, meaning that a post-hoc use of the US data may be a source of additional error due to lacking relevant information. 

Author Response

Reviwer 2 comments.This is a single-centre study, and all the results must be viewed as such – please stress this in the limitations section.

English will require moderate improvements.

Thank you very much for your comments.

We tried to improve our English sending the paper for extended editing (MDPI editing services)

The Abstract should not report univariate results, as they are misleading. On the contrary, you should report ORs and CIs for the multivariate significant predictors.

Univariate results were removed from the Abstract and only the multivariate analysis was commented.

The main question is, why bother with ultrasound if the operation can be performed? There is an additional cost, and the problem is only diagnosed. Of course, it may drive and guide more complicated cases, but it could be argued that this entire procedure is not needed in appendicitis.

About your question for the need or not of the preoperative US, our study wanted to examine if the preoperative US can distinguish the AUA from ACA. If this is possible the children with AUA could easily and effectively begin their treatment nonoperatively limiting the possibility of complications.

The sample breakdown is another problem – how many vases of appendicitis did you operate that did not have the pre-operative US?

15 patients with appendicitis didn’t have pre-operative US. However, we didn’t include these patients to our study

Change the less or equal P to 0.05 to only less than 0.05.

The p-value changed to lower than 0.05.

Downplay the univariate results; they are misleading.

Explain the ACA and AUA abbreviations again in the header of Table 1.

Explain what P stands for in Table 1, ACA vs all, AUA vs all or ACA vs AUA?

However, I checked the chi-square results, they are not the same as you showed them. Please verify.

Abbreviations ACA and AUA were explained in the header of Table 1.

p-value in Table 1 refers to the statistical tests performed to compare ACA vs AUA.

The p-value of chi-square test between hypervascularisation and histopathological classification was corrected from 0.195 to 0.197

 The discussion aligns well and restores some of my previous questions, so it fits with the paper scope nicely.

Please suggest more replication studies so these results can be verified and possibly used in the treatment algorithm in the future.

We suggested for further studies in the conclusion as you proposed

Please mention the fact that you did not use the live US, but a retrospective US result in the diagnostics; this can have a negative effect, as the US requires a lot of manipulation, meaning that a post-hoc use of the US data may be a source of additional error due to lacking relevant information. 

We mention that we had retrospective examine the patients US in the section of the limitations of our study. However our radiologists have specific protocol for the examination of the appendix  in children(instructions given from our pediatric radiologist)  , we referred this in the section of ultrasound examination

Round 2

Reviewer 2 Report

Agree